# Directing lateral growth of lithium dendrites in micro-compartmented anode arrays for safe lithium metal batteries

Peichao Zou[1], Yang Wang[1], Sum-Wai Chiang[1], Xuanyu Wang[1], Feiyu Kang[1,2] & Cheng Yang[1]

Uncontrolled growth of lithium dendrites during cycling has remained a challenging issue for lithium metal batteries. Thus far, various approaches have been proposed to delay or suppress dendrite growth, yet little attention has been paid to the solutions that can make batteries keep working when lithium dendrites are already extensively present. Here we develop an industry-adoptable technology to laterally direct the growth of lithium dendrites, where all dendrites are retained inside the compartmented copper current collector in a given limited cycling capacity. This featured electrode layout renders superior cycling stability (e.g., smoothly running for over 150 cycles at 0.5 mA cm$^{-2}$). Numerical simulations indicate that reduced dendritic stress and damage to the separator are achieved when the battery is abusively running over the ceiling capacity to generate protrusions. This study may contribute to a deeper comprehension of metal dendrites and provide a significant step towards ultimate safe batteries.

[1] Division of Energy and Environment, Graduate School at Shenzhen, Tsinghua University, Shenzhen, 518055, China. [2] School of Materials Science and Engineering, Tsinghua University, Beijing, 100084, China. Correspondence and requests for materials should be addressed to C.Y. (email: yang.cheng@sz.tsinghua.edu.cn)

Lithium (Li) metal can deliver the highest theoretical specific capacity (3,860 mA h g$^{-1}$) among all types of lithium battery anodes[1]. However, Li metal anode has the highest safety risk, which mainly derives from the uncontrolled growth of Li dendrites during repeated electrochemical charging/discharging, accompanied with dendrite-induced internal shorts, thus impeding the further commercialization of high-energy-density rechargeable lithium metal batteries (LMBs) such as Li-air and Li-sulfur batteries[2–5]. Efforts towards addressing this issue have been mainly focused on dendrite-growth-delay/suppression strategies via the employment of optimized electrolyte[6–10], modified separators[11–13], Li anode surface modifications[14,15], artificial anode surface coatings[16–19], accommodating nano- and micro-structured current collectors[20–25], etc. These achievements have provided elaborate insights into the feasibility of effective dendrite suppression via stabilization and homogenization of the solid-electrolyte-interface (SEI) layer or accommodation of electrodeposited Li metal and have promoted the commercialization of the metallic lithium anode in Li-metal-based secondary batteries. Whereas the extreme situation when controlled dendrite suppression/delay is lost in these strategies has been rarely discussed, because the emergence of Li dendrites cannot be completely avoided during prolonged cycling[3], especially when batteries are operated at high current densities, in overcharge ultimate, or at low operation temperatures[26,27]. On the other hand, the electrodeposition/dissolution behaviours of Li metal and the corresponding influencing factors are intrinsically complicated, rendering the control over these behaviours hardly predictable and extremely difficult to be managed with available technologies. This is one of the main reasons that the NEC Corporation finally decided to abandon metallic Li anodes in their commercial products, even though the NEC Corporation already had powerful technological solutions to largely inhibit dendrite growth in their battery products[28]. An ideal Li anode shall be dendrite free, yet in reality dendrite growth cannot be fully avoided. Therefore, an ultimately safe LMB solution may have to consider the presence of a number of dendrites, no matter when, where and how these dendrites grow from the anode. The relevant technological breakthrough could be the last line of defence to catastrophic battery failure.

In a sandwich cell structure, the electric field is predominately distributed in a vertical pattern; thus, upon Li dendrite formation, the dendrite will grow vertically towards the separator and cathode[3,4,29,30], and eventually could impale the separator and cause an internal short circuit in the battery[3]. Researchers have recently developed a series of Li anodes based on micro/nano-structured scaffolds[22,24,25,31], which enabled more even distribution of the electric field and reduced the effective current density by providing more electroactive sites as well as accommodating plated Li metal inside these 'host materials'. Thus, these strategies could homogenize the plating/stripping behaviour of Li on the electrode surface and eliminate the dendritic formation. However, the vertical growth phenomenon of Li metal towards separator still exists once plated lithium fills the electrode surface after a certain time[31]. For decades, intensive fundamental studies with well-established models have led to increasing understanding of the underlying growth mechanism of Li dendrites[3,4]. However, no systematic investigation or model has been proposed to controllably guide the growth of Li dendrites, especially from the view point of the designated electric field and at the level of the entire battery device.

Herein, we develop a scalable technology with photolithographic-level conformity for the fabrication of a poly-imide (PI)-clad copper grid current collector (E-Cu) for Li metal anodes, where the electric field exhibits a lateral pattern inside E-Cu and thus guides the Li dendrites to laterally grow within the interior Cu scaffold. Instead of suppressing/delaying dendritic growth, this technology allows the growth of Li dendrites, but modulates the growth to the direction parallel to the separator and thus the batteries can still safely operate even when dendrites massively exist. All the processes involved, including hot lamination, laser ablation and alkaline etching (as schematically illustrated in Fig. 1a), have been widely used in electronic and semiconductor industry for more than half a century[32–34], which can ensure the highest conformity level. In this anode structure, the designated hollow compartments serve as numerous small cages to absorb all Li dendrites and accommodate the volume change during Li plating/stripping, while the upper PI layer acts as a physical barrier to prevent Li dendrites from protruding out of the compartments, where the pinholes in the upper PI layer guarantee sufficient electrolyte diffusion. Experimental investigations combined with numerical simulations are performed to evaluate the merits of such a unique current collector for Li anodes over planar copper (P-Cu)-based electrodes in the cell configuration. By laterally directing the encapsulation of Li metal in the micro-compartments, the assembled LMBs consisting of E-Cu-based Li metal anode and LiFePO$_4$ (LFP) cathode can run for 250 cycles with a capacity retention of 100.7%.

## Results

**Fabrication and characterization of E-Cu**. The three-dimensional (3D) E-Cu was fabricated by a series of processes, including hot lamination, laser ablation and alkaline etching treatments, as schematically illustrated in Fig. 1a. After hot lamination, a P-Cu foil was sandwiched between two PI films. Then, pinholes arranged in a hexagonal pattern in the upper PI film were produced by laser ablation. After the final alkaline etching treatment, numerous hollow compartments inside the copper grid were obtained, which were coaxial with pinholes overhead to enable the most effective space utilization. A digital photograph of the final E-Cu current collector is displayed in Fig. 1b. Notably, either the pinholes in the upper PI film or the compartments inside the interior copper grid are the same in geometric size, owing to standard industrial fabrication processes as is addressed in detail in the experimental section. Digital images of the current collector samples before and after different processing steps are displayed in Supplementary Fig. 1.

Typical scanning electron microscopy (SEM) images of the copper scaffold layer are displayed in Fig. 1c,d (overview) and Fig. 1e (cross-section-view), from which we can observe a homogenous distribution of the cylindrical compartments with uniform diameter. By tailoring the fabrication parameters during laser ablation and the etching treatment, the compartment diameter can be accurately adjusted from 120 to 180 µm, as depicted in Supplementary Fig. 2. In the present work, E-Cu with a diameter of 150 µm in the interior compartments and 45 µm in surface pinholes of the upper PI film was chosen to investigate its application in LMBs as an anode current collector unless otherwise noted; the P-Cu current collector was used as control. The E-Cu in such a geometric size has an electroactive area ratio of ~1.06 (see Supplementary Table 1), which is very close to that of P-Cu (1.0). In other words, the exposed area of conductive copper in E-Cu is similar to that of P-Cu when the surface areas of these two electrodes are the same. Detailed information about the electroactive area ratio of E-Cu is given in the supplementary information.

As the volume of the compartments determines the maximum capacity of E-Cu for Li storage (namely, the capacity density of the anode)[22], mercury porosimetry analysis combined with theoretical calculations were conducted to evaluate the accommodation capacity of E-Cu. The volume density of the hollow

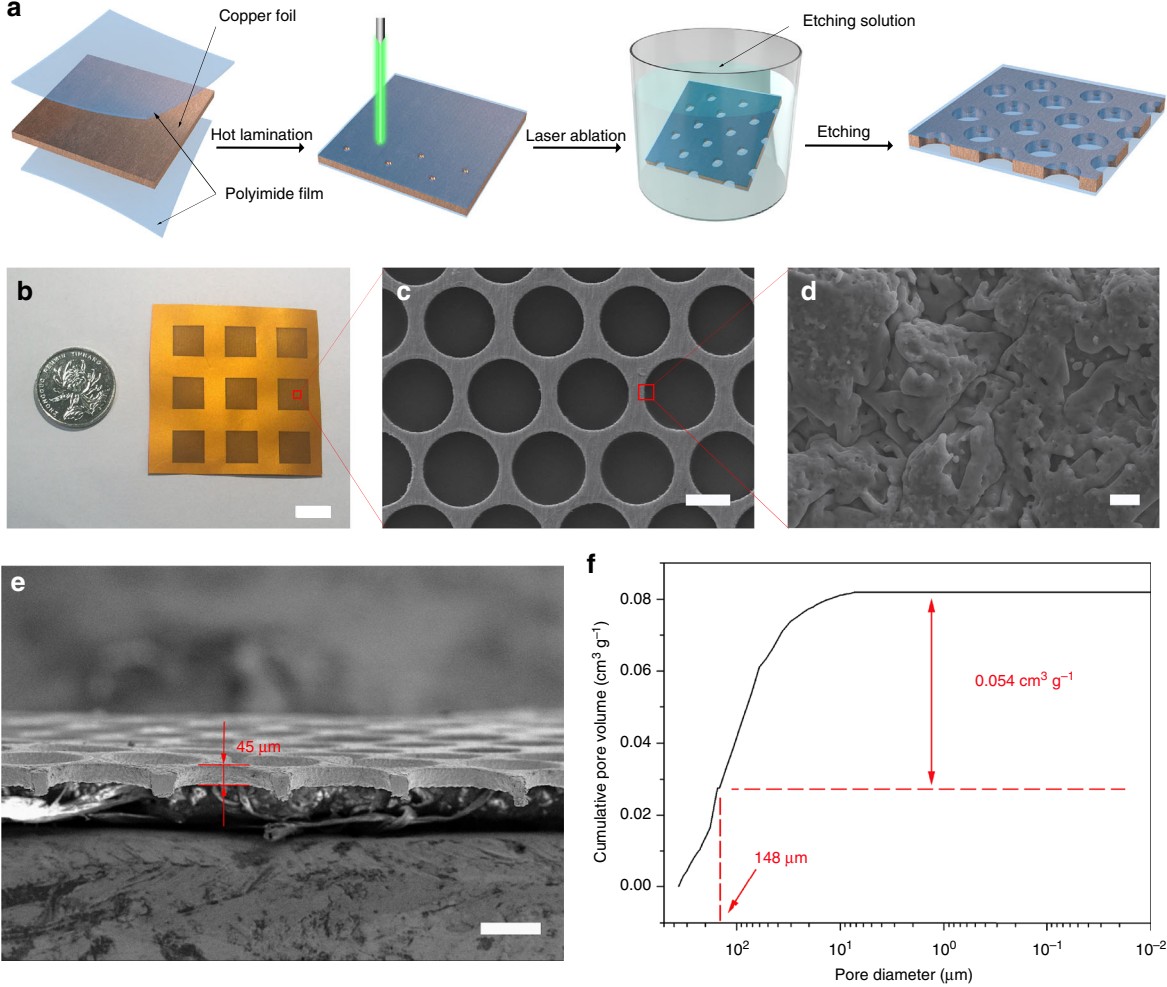

**Fig. 1** Preparation and characterization of E-Cu. **a** Schematic of the fabrication process of E-Cu. **b** Photograph of the as-obtained E-Cu current collector. **c** Typical SEM overview image of E-Cu after peeling off the coated PI films. **d** Magnified SEM image of the edge of the copper scaffold from **c**. Scale bars in **b**–**d** are 1 cm, 100 μm and 1 μm, respectively. **e** Typical cross-sectional SEM image of E-Cu after peeling off the surface PI films. Scale bar in **e** is 100 μm. **f** Cumulative pore volume of E-Cu by mercury porosimetry. Herein, the diameter of interior compartments is controlled to be ~150 μm, which was in accordance to the porosity analysis result (148 μm), as marked in red colour in **f**. The weight of E-Cu tested here was 0.3141 g and the total surface area for all the compartments was 9 cm$^2$; thus, the effective pore volume of E-Cu is $1.88 \times 10^{-3}$ cm$^3$ cm$^{-2}$

compartments for E-Cu was calculated to be $\sim 1.98 \times 10^{-3}$ cm$^3$ cm$^{-2}$ (see Supplementary Table 1), agreeing well with the mercury porosimetry analysis result ($1.88 \times 10^{-3}$ cm$^3$ cm$^{-2}$, Fig. 1f). Then, the theoretical areal capacity density of the Li anode based on E-Cu with fully filled Li metal was estimated to be 4.1 mA h cm$^{-2}$, which can cater to most of the energy density demands for energy storage system[22]. By increasing the thickness of the copper foil interlayer, the capacity for Li storage of E-Cu can be further enhanced.

**Simulation of the electric field distribution and lithium plating/stripping behaviours in E-Cu.** The distribution of the electric field generated in P-Cu and E-Cu is schematically depicted in Fig. 2a. On P-Cu, the direction of electric field exhibits a simple vertical pattern (perpendicular to the separator). However, the distribution of the electric field generated inside E-Cu presents a unique lateral pattern, confirmed by a numerical simulation using an electrical conduction model as exhibited in Fig. 2b; in this pattern, the electric field propagates from the counter electrode, through the pinhole, and extends laterally to the Cu scaffold surface. This unique distribution pattern derives from the

distortion effect of the top insulative PI layer on the electric field. Here, the electric field distribution is considered as a dominant factor (see Supplementary Information) to modulate the growth behaviour of lithium dendrites[35], as discussed below.

Along the electric field distribution, plated Li metal primarily forms into small and mossy Li dendrites on the smooth P-Cu surface due to the limited number of electroactive sites. During the charging process, the subsequent dissolution of Li will result in many sharp ends and dead Li on the P-Cu surface. As Li metal is preferentially deposited along the sharp ends where the local current density is dramatically increased[5], larger Li dendrites and more dead Li will be evolved after repeated cycles (Fig. 2c). In contrast, owing to the existence of the insulative PI film on E-Cu, Li metal is limited to laterally deposit inside the Cu scaffold and grows into Li dendrites. Even after prolonged cycling, Li dendrites will always be confined inside these hollow compartments as long as the cycling capacity is not exceeded (Fig. 2c). Here, the upper PI film in E-Cu can act as a physical barrier that shields Li dendrites from protruding out of the pinholes in upper PI film. Despite some distortion effect on the electric field distribution within the compartment with the presence of this PI layer, the electrochemical plating/stripping behaviour of Li metal in E-Cu

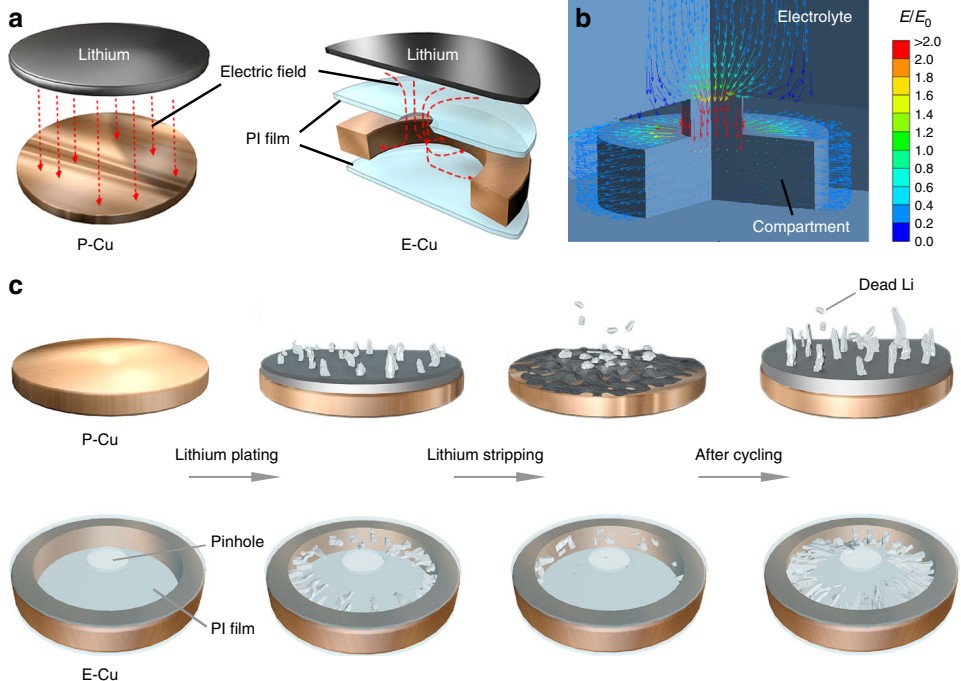

**Fig. 2** Schematic electrochemical plating/stripping behaviours of Li metal anodes. **a** Schematic illustration of the electric field distribution in P-Cu and E-Cu. **b** Simulated result of the electric field distribution in E-Cu. **c** Schematic illustration of the evolution of different Li anode structures based on P-Cu and E-Cu during Li plating/stripping

can still be observed, as the electric field can propagate into the compartment (see Supplementary Figs. 3 and 4 for discussion).

Once the cycling capacity approaches the limit capacity of E-Cu for effective Li storage, vertical Li dendrites will protrude out of the pinholes of the upper PI film (extreme case). Nevertheless, these protrusions are often long and curvy compared with the short and sharp protrusions on P-Cu, indicating that these dendrites are mechanically much weaker than those on P-Cu, and therefore producing lower stress and being less probable to impale the separator. Accordingly, we simulated this stress using a structural stress analysis model and observed an ~60% reduced stress from the protruded Li dendrites in E-Cu as compared with that in P-Cu (Fig. 3). In these simulations, a vertical dendritic protrusion is positioned against the separator under different deposition capacities (Fig. 3a). When the compartment is filled to the top PI film with Li dendrites (Fig. 3c–e), the stress on the separator is still significantly less than that of the control case (planar configuration, Fig. 3b). In other words, the predicted reduction of the protrusion stress will further alleviate the safety problem of lithium anodes based on E-Cu and enhance the structural integrity. Notably, broken Li strips will remain inside the compartment and thus Li metal can only grow along these broken strips until these strips are reconnected with newly formed lateral Li dendrite, which is quite different to the situation for P-Cu; in this case, the stress distribution in reconnected dendrites would be similar to the case without remnants.

To investigate the morphological evolution of Li metal in more detail, 0.5, 1 and 2 mA h cm$^{-2}$ of Li metal were separately deposited on E-Cu and P-Cu at the same electrode current density of 0.5 mA cm$^{-2}$. From the ex situ SEM images shown in Fig. 4a–f, Li dendrites laterally grew up and gradually filled the compartments with an increasing amount of plated Li. Generally, the deposited Li dendrites presented a typical whisker shape (Fig. 4d–f), which can be ascribed to the inducement of the designated electric field as discussed above. In addition, the plated Li dendrites were caged inside the hollow compartments without

protruding upwards from the upper pinholes in PI films; as shown in Supplementary Fig. 5, no obvious Li metal was found protruding out of the pinholes in PI films. When depositing 2 mA h cm$^{-2}$ of Li metal into E-Cu at a higher current density of 1 mA cm$^{-2}$, a similar morphology for the Li dendrites was also observed (see Supplementary Fig. 6). In contrast, Li metal mainly demonstrated a sharp and short spine shape on P-Cu (see Supplementary Fig. 7), which became larger and thicker with more deposited Li metal, in good accordance to the theoretically expected results (Fig. 2c) and previous reported results[31,36]. In addition, an increasing amount of 4 mA h cm$^{-2}$ of Li metal was further plated into E-Cu to analyse the storage accommodation of Li metal, where the deposition capacity almost approaches the theoretical limit capacity for Li storage (4.1 mA h cm$^{-2}$, Supplementary Table 1) in E-Cu. As displayed in Supplementary Fig. 8, Li metal filled the compartments in a more compact manner. However, in this case, few protruded Li metal was observed after the first deposition process. This result was due to the plated Li metal in the upper section of compartments shielding the further growth of Li metal in the bottom section, thus making the deposited Li metal not completely compact.

Despite the long-whisker shape, the plated Li metal inside E-Cu can also be reversibly stripped. When 1 mA h cm$^{-2}$ of Li metal was stripped from the Li anode before 2 mA h cm$^{-2}$ of Li metal was deposited into E-Cu first, the residual Li metal aggregated around the copper skeleton (Fig. 4d), indicating that the dissolution behaviour started from the heads of the Li whiskers, which are the nearest sites to the separator and exhibit the strongest electric field along the whiskers (Supplementary Fig. 4). Moreover, Li metal still grew into long whiskers without obvious protruded Li metal when another 1.0 mA h cm$^{-2}$ of Li metal was plated back into E-Cu (Fig. 4e). As displayed in Fig. 4f, the plated Li metal inside the compartment can be almost completely dissolved when the battery was recharged to a cut-off voltage of 0.5 V. The stripping behaviour of Li metal on P-Cu is displayed and discussed in Supplementary Fig. 9.

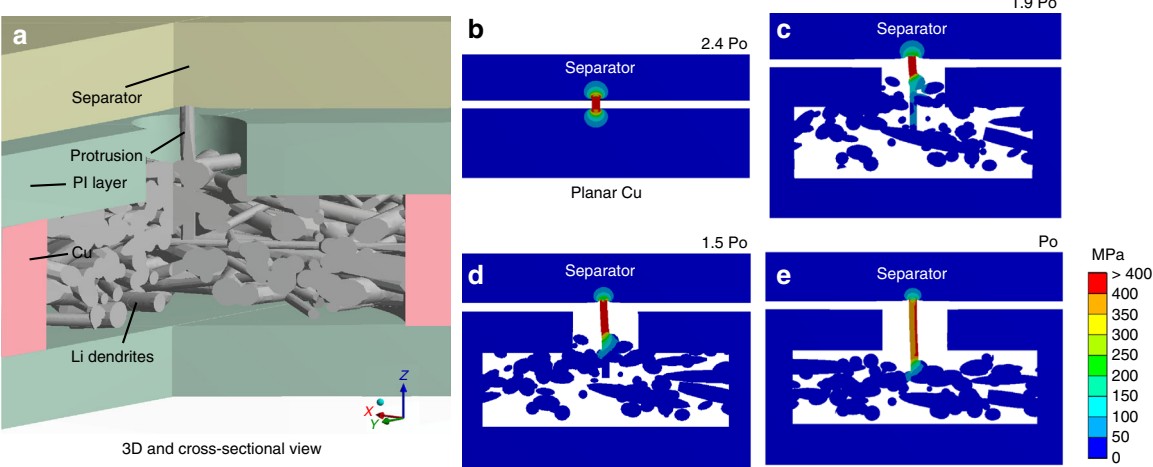

**Fig. 3** Simulation of the von Mises stress distribution of the dendritic protrusions at different deposition heights. A vertical protrusion is grown from the lower deposition to the separator in all cases. This protrusion is located at the center of the *XY* plane. **a** Three-dimensional and cross-sectional view of the E-Cu@Li model. **b** Distribution of the protrusion stress in the control case with two-plate electrodes. **c–e** Distribution of the protrusion stress in the E-Cu-based case with the dendrite layer reaching **c** 95%, **d** 75% and **e** 50% of the height of the compartment. In the top right-hand corners of Fig. 3b–e, the relative area-average von Mises stress from the protrusion towards the separator is indicated

To further evaluate the cycling performance and structure stability, we conducted a repeated charging/discharging cycling test by plating/stripping 2.0 mA h cm$^{-2}$ of Li metal at 0.5 mA cm$^{-2}$ on E-Cu- and P-Cu-based batteries over 100 cycles. The tested symmetric cells based on E-Cu and P-Cu are schematically depicted in Fig. 5a. After prolonged cycling, plated Li metal evolved into agglomerated particles rather than remaining in a pristine whisker shape in E-Cu (Fig. 5d). Even so, no dramatic dendrite protrusions were found out of the structure, as can be seen in Fig. 5b. Once the Li particles form, dendritic growth will be sequentially impeded; consequently, dendritic protrusions will be further inhibited. Thereby, operational safety over prolonged cycling for the E-Cu-based anode is expected, as Li metal is still stored inside for the entire time. On the other hand, the plated Li dendrites on P-Cu after repeated cycles became larger and thicker, still exhibiting an uneven surface (Fig. 5c). These sharp and rigid dendrites can directly contact with the separator, resulting in a higher occurrence of internal shorts.

**Electrochemical performance evaluation of E-Cu.** In order to evaluate the safety performance, we analysed the plating/stripping behaviour of Li metal as well as the cycling stabilities on Li electrodes with E-Cu and P-Cu current collectors tested in a half-cell configuration (E-Cu//Li and P-Cu//Li) by comparing their Coulombic efficiencies (CEs), discharge/charge voltage profiles and average voltage hysteresis. Here, 0.5, 1.0 and 2.0 mA h cm$^{-2}$ of Li metal was separately plated onto these current collectors, followed by recharging to a cut-off voltage of 0.5 V at the same current density of 0.5 mA cm$^{-2}$ in each cycle. As shown in Fig. 6, the CE of the E-Cu//Li battery remained ~99% with a cycling capacity of 0.5 mA h cm$^{-2}$ and ~90% with a cycling capacity of 1 mA h cm$^{-2}$ after 150 cycles. In comparison, the P-Cu//Li battery achieved a fluctuant CE of only 92% with a plating/stripping capacity of 0.5 mA h cm$^{-2}$ after 150 cycles. When depositing 1 mA h cm$^{-2}$ of Li metal onto P-Cu, the P-Cu-based cell exhibited an obvious and gradual decrease in CE, dropping below 70% after 80 cycles and eventually came to short circuit after < 90 cycles because of the unstable structure of Li dendrites and the SEI layer[16]. At a higher cycling capacity of 2 mA h cm$^{-2}$, the P-Cu-based Li metal anode showed poorer cycling performance with an initial short circuit after less than 50 cycles, whereas the E-Cu-

based anodes could still run for 130 cycles without failure. In addition, the E-Cu-based anode still presented superior cycling performance when cycling at a plating capacity of 4 mA h cm$^{-2}$, which could run for more than 50 cycles until a final short circuit (see Supplementary Fig. 10). Similar results can also be obtained by analyzing the discharging/charging voltage-time profiles and average voltage hysteresis of the Li electrode plating/stripping with 1 mA h cm$^{-2}$ of Li metal at 0.5 mA cm$^{-2}$ (the stripping process was controlled by the cutoff voltage), as shown in Supplementary Fig. 11.

Figure 6b displays the voltage-time profiles of Li metal on E-Cu and P-Cu at a constant cycling current density of 0.2 mA cm$^{-2}$ within 500 h (the stripping process was controlled by the areal capacity of Li metal), from which a stable charging/discharging voltage plateau was observed for the E-Cu-based electrode and a fluctuant one for the P-Cu-based electrode. As shown in Fig. 6c, the voltage hysteresis of Li plating/stripping in the 3D E-Cu stabilized at approximately 100 mV even after 100 cycles, indicating the excellent cycling stability. In contrast, the voltage hysteresis of Li plating/stripping on P-Cu delivered random voltage oscillations, which were mainly derived from the instability of Li/electrolyte interface[16]. In addition, several dendrites-induced short circuits occurred in the P-Cu-based Li anode during cycling, which can be observed from abrupt voltage drops from high potentials to lower potentials in the voltage profile, as indicated in Fig. 6c.

Aiming to further verify the potential of the 3D E-Cu current collector for practical application, full cells were assembled using Li anodes with the E-Cu and P-Cu current collectors against commercial LFP cathodes (defined as E-Cu@Li//LFP and P-Cu@Li//LFP, respectively), and their corresponding electrochemical performances are shown in Fig. 6d. The initial charging/discharging curves of different full cells at 1 C are shown in Supplementary Fig. 12, from which clear charging/discharging potential plateaus are observed, indicating a reversible cycling process. Furthermore, the E-Cu@Li//LFP cell delivers an initial discharge capacity of 130.2 mA h g$^{-1}$ with a CE of 93.7% and the P-Cu@Li//LFP cell presents a similar result (138.3 mA h g$^{-1}$ and 93.5%, respectively). However, the E-Cu@Li//LFP cell can still deliver a high discharge capacity of 131.1 mA h g$^{-1}$ with a CE of 99.5% after 250 cycles at 1 C. In comparison, the capacity retention of the P-Cu@Li//LFP cell is only 58.6% after 250 cycles.

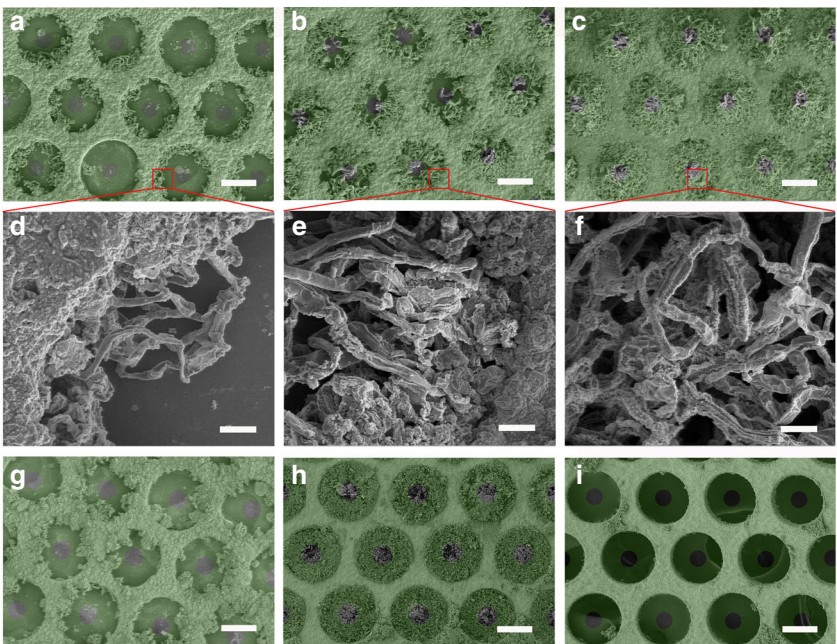

**Fig. 4** Morphologies of Li metal anodes during plating/stripping. **a–f** SEM images of E-Cu after Li deposition at 0.5 mA cm$^{-2}$ with different capacities: **a**, **d** 0.5 mA h cm$^{-2}$; **b**, **e** 1 mA h cm$^{-2}$; and **c**, **f** 2 mA h cm$^{-2}$. **g–i** SEM images of E-Cu after **g** stripping away 1 mA h cm$^{-2}$ of Li from E-Cu, followed by **h** plating back 1 mA h cm$^{-2}$ of Li into E-Cu and **i** re-stripping all Li metal from E-Cu. In these cases, 2 mA h cm$^{-2}$ of Li was primarily deposited into E-Cu at 0.5 mA cm$^{-2}$. The green colour in the graphs of **a–c** and **g–i** represents the upper PI film on the Cu scaffold. Scale bars in **a–c** and **g–i** are 100 μm and in **d–f** are 10 μm

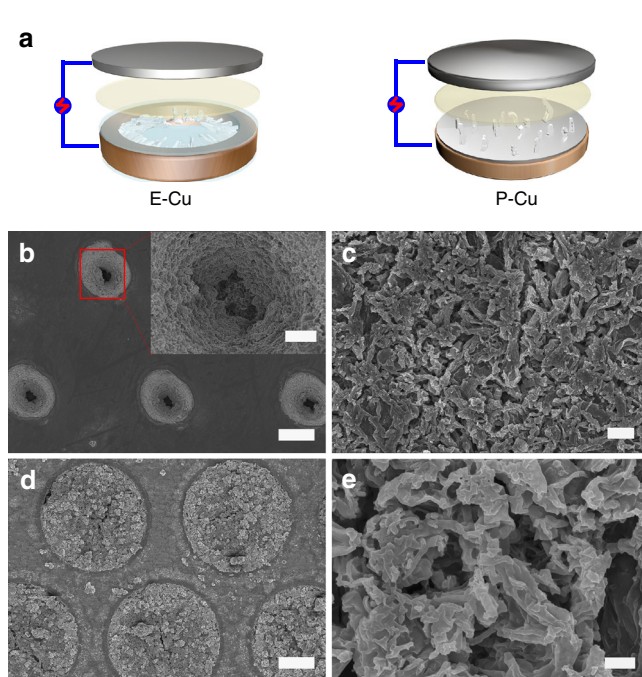

**Fig. 5** Morphologies of Li metal anodes after cycling. **a** Schematic illustration of symmetric cell structure consisting of Li metal electrodes based on E-Cu and P-Cu with a Li metal foil counter electrode. **b** Surface top-view SEM image of E-Cu after cycling for 150 cycles at 0.5 mA cm$^{-2}$; the inset is the magnified image of the edge of the pinholes in the PI film from the selected area in **a**. **c** Overview SEM image of P-Cu after cycling. **d** Overview SEM image of E-Cu after peeling off the upper PI film. **e** Magnified overview SEM image of P-Cu after cycling. The scale bars in **b**, **c**, **d** and **e** are 50 μm, 2 μm, 50 μm and 500 nm, respectively. The scale bar in the inset of **b** is 10 μm

These results demonstrate that the E-Cu-based LMBs display a similar initial capacity and CE but enhanced cycling performance when compared to the conventional planar Cu-based LMBs.

## Discussion

In summary, this manuscript demonstrates that guiding Li dendrites to laterally grow in compartmented micro-electrodes is an effective way to manage the safety issue of Li anodes, which is different to the widely adopted strategies of suppressing/delaying dendritic growth. The model structure can effectively change the electric field distribution and accommodate the plated Li metal. Experimental results reveal that Li metal can be continuously accommodated inside the hollow compartments of E-Cu without protruding out of the upper PI film even after long-term cycling by controlling the deposition capacity. Simulation analysis further confirms the feasibility of such technology in managing the risk of dendrite growth and reducing the dendrite protruding stress towards the separator by 60%. Owing to the confinement effect, the E-Cu-based Li anode exhibits a superior cycle life and a more stable voltage hysteresis to the P-Cu-based Li anode in both Cu//Li systems and Cu@Li//LFP batteries.

As E-Cu studied in the present work has a similar electroactive area ratio as P-Cu (close to 1), we can deduce that if the interval distance between two adjacent compartments is reduced and the volume ratio of the compartments is increased, the electroactive area ratio can be further improved and thus a better comprehensive performance for Li anodes based on E-Cu can be obtained[22]. This technology is compatible with most state-of-the-art dendrite growth delay/suppression technologies and can be further improved with the development of microfabrication and computational techniques. Besides, the excellent compatibility between this compartmented electrode structure and industrially available fabrication techniques, including hot lamination, laser ablation and alkaline etching, also renders this technology with unprecedented conformity and reliability. Therefore, this technology would be a critical step towards large-scale manufacturing

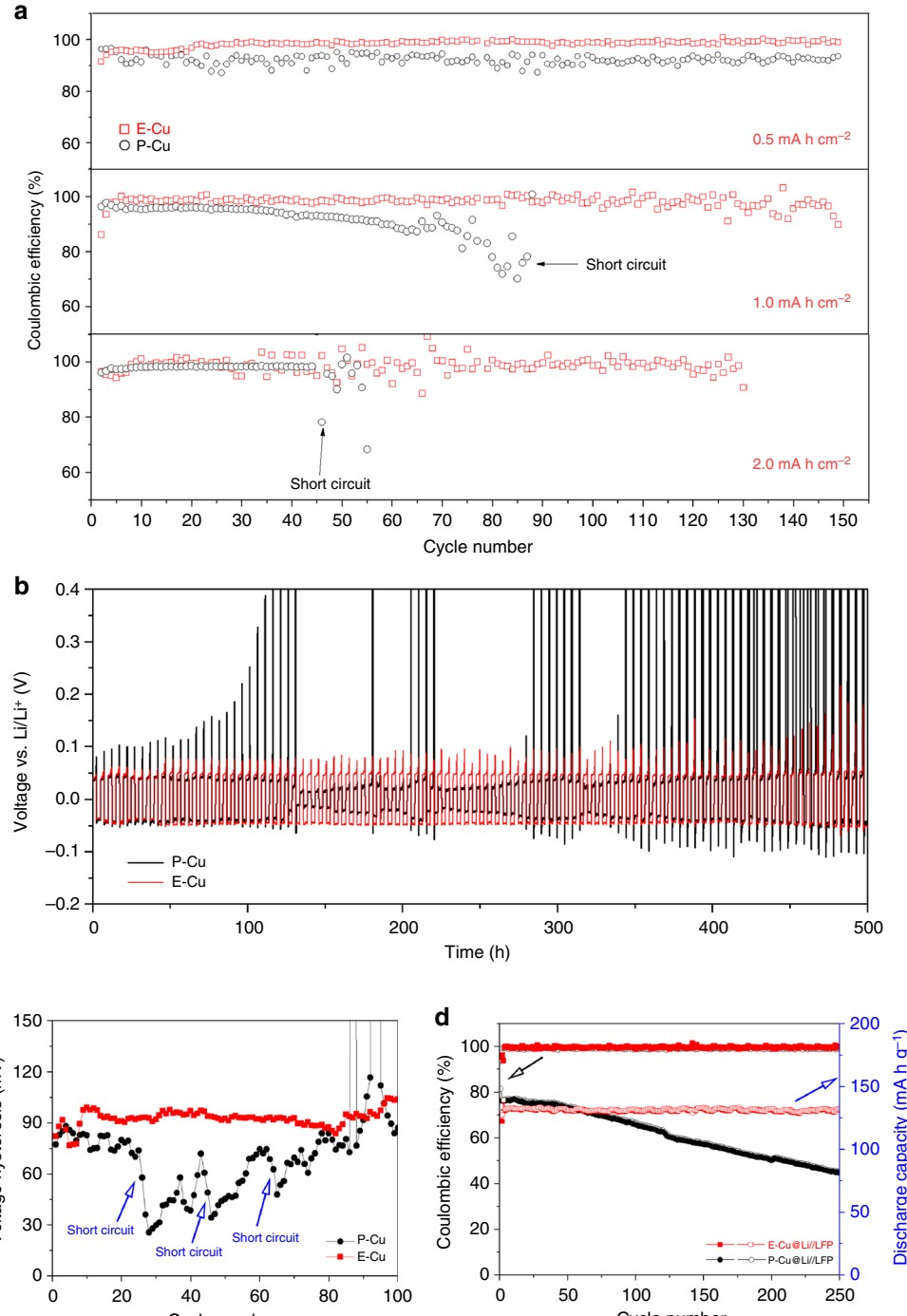

**Fig. 6** Electrochemical performance of Li metal anodes. **a** Comparison of cycling performances (Coulombic efficiency) of the P-Cu- and E-Cu-based Li metal anodes at 0.5 mA cm$^{-2}$ with varied amounts of plating/stripping Li metal. **b** Voltage-time profiles and **c** average voltage hysteresis of Li metal anodes in Cu@Li//Li systems based on E-Cu and P-Cu with a plating/stripping capacity of 0.5 mA h cm$^{-2}$ at 0.2 mA cm$^{-2}$. In this case, 2 mA h cm$^{-2}$ of Li was plated at 0.2 mA cm$^{-2}$ in two current collectors. **d** Cycling performances of the Cu@Li//LiFePO$_4$ battery systems using Li metal anodes with E-Cu and P-Cu at 1C

of Li metal anode based batteries. This unique strategy is a first attempt to deal with the extreme situation when lithium dendrites have massively presented via manipulating the electric field distribution and growth dynamics of Li dendrites, which provides new insights into the unwelcomed dendrite growth issue, and will inspire technological developments of other metal anodes in rechargeable systems.

## Methods

**Preparation of 3D PI-clad copper current collectors.** Primarily, two PI films with a thickness of 25 μm were separately coated onto the upper and bottom surface of the planar Cu foil (45 μm in thickness) via hot lamination at 180 °C. Subsequently, laser ablation processing was employed to produce a pinhole array (aligning in a hexagonal pattern; diameter of 45 μm) in the upper PI film with a nanosecond pulsed laser (EP-15-DW, Han's Laser Technology Co., Ltd, China). Finally, the PI-clad copper foil was immersed into an alkaline etching solution to allow the etching of the copper interlayer and the extension of the compartment array to obtain the

eventual current collector. Herein, the etching solution was prepared by adding ammonia water into a pristine mixed aqueous solution (Alfa Aesar) containing 1 M $CuCl_2$ (Alfa Aesar) and 0.5 M $NH_4Cl$ (Alfa Aesar) until the pH value of the mixed solution was tuned to 8 with a pH meter (F-71, Horiba, Co., Ltd, Japan). Copper current collectors were immersed into ethanol solution with ultrasonic treatment to remove the residual carbon resulting from the laser ablation process and etching solution before and after the etching treatment. The primary chemical reactions involved in the preparation of the etching solution and the etching process of interior copper can be expressed as follows:

$$CuCl_2 + 4NH_3 \rightarrow Cu(NH_3)_4Cl_2 \quad (1)$$

$$Cu(NH_3)_4Cl_2 + Cu \rightarrow 2Cu(NH_3)_2Cl \quad (2)$$

It is worth stressing that the etched solution can be self-renewed with self-oxidation behaviour in air, thereby rendering the fabrication process much more environmentally friendly. The main chemical reaction occurring in the regeneration of the etched solution is:

$$2Cu(NH_3)_2Cl + 2NH_4Cl + 2NH_3 + 1/2O_2 \rightarrow 2Cu(NH_3)_4Cl_2 + H_2O \quad (3)$$

**Characterization**. Optical microscopic images of E-Cu with compartment array in a varied diameter after alkaline etching treatment were taken on a metallographic microscope (Olympus GX 51). The microscopic morphologies of E-Cu and Li metal after designated electrochemical tests in different current collectors were characterized by field emission SEM (ZEISS SUPRA 55, 5 kV, Germany). To be noted, the SEM samples related to Li metal were prepared by disassembling tested cells in an argon-filled glove box and then the extracted Li anodes were rinsed with dimethyl carbonate (DMC, Alfa Aesar) and vacuum dried at 60 °C for 2 h to remove the residual solvent. Before SEM analysis, the samples were transferred into a vacuum box for short-time storage.

**Electrochemical measurements**. All electrochemical measurements were conducted in a CR2032-type coin cell set-up on a LAND 2001A electrochemical testing system at 25 °C, where all cells were assembled or disassembled in an argon-filled glove box. The electrolyte used in the Cu//Li batteries was 1 M lithium bis(tri-fluoromethanesulfonyl)imide) in a cosolvent of 1,3-dioxolane and dimethoxy-ethane (1:1 in volume, 40 μl) with 1% $LiNO_3$ additive, where $LiNO_3$ is expected to contribute to the formation of a stable SEI layer on the Li anodes[37]. The CEs for the Cu//Li LMBs were tested at 0.5 mA cm$^{-2}$ with 0.5, 1.0, 2.0 or 4.0 mA h cm$^{-2}$ of plated/stripped Li metal, where E-Cu or P-Cu was employed as the cathode, Li foil as the anode and a commercial microporous membrane (Celgard 2400) was utilized as the separator during the measurements. In order to stabilize the SEI layer and remove surface contaminations, the E-Cu//Li and P-Cu//Li batteries were primarily charged and discharged between 0 V and 1 V (vs. Li$^+$/Li) at 50 μA for five cycles[19,22]. The discharging/charging voltage-time profile in this period is exhibited in Supplementary Fig. 13. Afterwards, 0.5, 1.0, 2.0 and 4.0 mA h cm$^{-2}$ of Li metal were separately plated onto the current collector at a current density of 0.5 mA cm$^{-2}$ and then the current collector was charged to a cut-off voltage of 0.5 V (vs. Li$^+$/Li) to strip the deposited Li metal at the same current density of 0.5 mA cm$^{-2}$ for each cycle.

The Cu@Li//LFP full cells were comprised of LFP electrode as the cathode, 2 mA cm$^{-2}$ of deposited Li metal on E-Cu or P-Cu as the anode and a microporous membrane as the separator (Celgard 2500). The electrolyte used in this system consisted of 1 M $LiPF_6$ in a nonaqueous solution of EC/EMC/DMC (in a volume ratio of 1:1:1, Dongguan Shanshan Battery Materials Co., Ltd). The LFP electrode was prepared by mixing commercial LFP powder (Shenzhen Dynanonic Co., Ltd) with carbon black (Super-P) and polyvinylidene difluoride (Alfa Aesar) in a fixed ratio of 8:1:1 in N-methyl-2-pyrrolidone (Aldrich) solvent dispersant, followed by coating this slurry onto a piece of aluminium foil and a vacuum drying at 110 °C overnight. For the anodes, 3D E-Cu or P-Cu was first assembled into a half cell with Li foil as the counter electrode, followed by plating 2 mA h cm$^{-2}$ of Li metal into these current collectors at 0.5 mA cm$^{-2}$. Then, the Li anodes were extracted from the above cells and subsequently rinsed with DMC and dried in a vacuum oven at 60 °C for 2 h to remove the residual solvent. Afterwards, the extracted Li anodes were reassembled into full cells against the LFP electrode. The integrated Cu@Li//LFP cells were cycled at 1 C at 2.4–4.2 V (vs. Li$^+$/Li) to investigate the cycling performance, before which the assembled cells were activated at 0.1 C for five cycles.

**Numerical simulation**. Finite element analysis (FEA) simulations were performed on a bare E-Cu model and E-Cu@Li model to predict the compartment effects on the electric field distribution as well as the lithium dendrites growth inside the compartments. The E-Cu@Li model was established on a single compartment with cylindrical dendrites inside, which were randomly generated using the Monte Carlos method. The FEA package from ANSYS Inc. was used for the simulations and post processing.

**Data availability**. The data that support the findings of this study are available from the corresponding author upon request.

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

## Acknowledgements

This work is supported by the National Key Basic Research Program of China (Project No. 2014CB932400), National Nature Science Foundation of China (Project No. 51607102 and 51578310), China Postdoctoral Science Foundation (Project No. 2016M601017 and 2016M601001) and Guangdong Province Science and Technology Department (Project No. 2015A030306010, 2014A010105002 and 2015B010127009).

## Author contributions

P.Z., Y.W. and C.Y. conceived and designed the experiments. P.Z. and Y.W. performed the major experiments. S.-W.C. performed the simulation study. P.Z., Y.W. and X.W. aggregated the figures. P.Z. wrote the draft. C.Y. revised the manuscript. All authors discussed the results and commented on the manuscript.

## Additional information

**Competing interests:** The authors declare no competing financial interests.

