## [Peer Review File · Nature Communications]

Reviewers' comments:

Reviewer #1 (Remarks to the Author):

The manuscript demonstrates a novel anode design for Li batteries to suppress safety/degradation effects of dendrite growth. The research presents an interesting alternative to typical dendrite suppression approaches.

The experimental aspects of the research are thoroughly detailed and explained. The computational model is not explained as thoroughly. The details of the model are lacking and no reference are provided to find those details. The connection between the modeling and experiments is lacking in the main paper and only discussed in the supplemental section. The manuscript would be improved by including details on the governing equations of the model and details of the Ansys models used.

The manuscript contains a number of grammar error and many sentences are written in poor English making some statements hard to understand.

Reviewer #2 (Remarks to the Author):

This paper presents evidence that a new technique that is described here will inhibit dendrites from growing through the separator, shorting the cell. The paper presents evidence for the efficacy of this technique. This is an extremely nice piece of work.

However, I do not recommend publishing in Nature Communications unless/until the authors compare their results to other, equally efficacious techniques and demonstrate that their technique offers benefits compared to them. One example to compare to is from Cui, attached.

Otherwise, I don't understand the electric field calculations. They appear to have been done assuming no electrolyte, which is unrealistic. In fact, much of the potential gradients will occur through the electric double layer. Is there something I am missing?

Because this paper presents a wealth of interesting data and analysis, it would be *my personal preference* to see a longer version of this paper. For example, it would be nice if some of the 16 Supplementary Figures could appear in the text, where more people would see them.

RESPONSE TO THE REVIEWER'S COMMENTS AND QUESTIONS:

We appreciate the reviewers for all the constructive questions and comments. Detailed responses are as follows.

Reviewer #1 (Remarks to the Author)

(1) The experimental aspects of the research are thoroughly detailed and explained. The computational model is not explained as thoroughly. The details of the model are lacking and no reference are provided to find those details.

Response: Thanks for the comment. Explicit and detailed explanations of the computational model are very important in elucidating the key points to readers. In the revised supporting information, we have updated the discussion section about the computational model setup and the related figures in a more clarified manner. In addition, a few more related references are added as well.

Original text 1:

Simulation setup: A finite element analysis method (FEA) simulation has been performed to predict the compartment effects on the distribution of electric field as well as the growth of lithium dendrites inside the compartments of E-Cu. As for E-Cu, the compartments have cylindrical structure with diameter (D) of 150 μm and height (H) of 45 μm , and the thickness of the upper PI film is 25 μm . The maximum distance for Li ion diffusion in vertical direction (x) is assumed to be 100 μm . In this work, all numerical simulation analysis about E-Cu is conducted on one compartment (Fig. 2b), which is the repeat unit of E-Cu and thus is expected to reflect the holistic phenomena occurred in all compartments of E-Cu. Then, we build the E-Cu@Li structure model based on one compartment with randomly distributed cylindrical lithium dendrites, where each of these dendrites randomly starts from the Cu scaffold and grows randomly and vertically inside the compartment. The same pseudo-random generator is used for all simulation cases, so as to reduce the random noise and ensure a fair comparison. According to the SEM observations (Fig. 3), the diameter distribution of the dendrites is estimated to range from 3 to 10 μm , while the length distribution is from 0 to 120 μm . The total number of dendrites depends on the volume of plated lithium in the compartment, which varies from 0% to 60% in the present simulated cases. In addition, typical conductivity and mechanical strength values are assumed for Li metal, PI film, Cu, electrolyte, and separator. An exception is that the conductivity of the Li dendrites is considered to be half of normal Li metal due to the existence of crystal boundaries in the dendrites; even so, it is still much better than that of the PI membrane and electrolyte. When a potential difference of V_0 (V) is applied across the top and bottom electrodes, we can obtain the electric field intensity of $E_0 = V_0/100$ ($\text{V } \mu\text{m}^{-1}$) generated across these two electrodes. Then, we use this E_0

to normalize the values of corresponding E field results as discussed below.

Revised text 1:

Simulation setup: A finite element analysis (FEA) method simulation was performed to predict the compartment effects on the distribution of electric field (E field) as well as the growth of lithium dendrites inside the compartments of E-Cu. The ANSYS models used are electrical conduction model^{1, 2} and structural stress analysis model^{3, 4} (based on the Hook's law). In this work, all numerical simulation analysis about E-Cu is conducted on one compartment (Fig. 3, top and side views), which is a typical unit of E-Cu and thus is expected to reflect the average phenomena occurred in all other compartments.

As for bare E-Cu model (see Supplementary Fig. 3b), the compartment has a cylindrical structure with the diameter (D) of 150 μm and height (H) of 45 μm , and the thickness of the upper PI film is 25 μm . The height of the bulk electrolyte in the domain is assumed to be 100 μm . Typical conductivity and mechanical strength are assumed for Li metal, PI film, Cu, electrolyte, and separator, as shown in Supplementary Table 2. The conductivity of the Li dendrites is considered to be half of normal Li metal due to the existence of crystal boundaries in the dendrites; even so, it is still much better than that of the PI membrane and electrolyte.

Then, we built the E-Cu@Li structure model based on one compartment with randomly distributed cylindrical lithium dendrites via Monte Carlos method with random generator (see Fig. 3a), where each of these dendrites randomly starts from the Cu scaffold and extends into the compartment. The diameter and length of each cylindrical dendrite are generated as a white noise distribution with the following ranges: according to the SEM observations (Fig. 4), the diameter distribution of the dendrites is estimated to range from 3 to 10 μm , while the length distribution is from 0 to 120 μm . The total number of dendrites depends on the volume of plated lithium in the compartments, which varies from 0% to 60% in the present simulated cases. When a potential difference of V_0 (V) is applied across the top and bottom electrodes, we can obtain the electric field intensity of $E_0 = V_0/100$ ($\text{V } \mu\text{m}^{-1}$) generated across these two electrodes. Then, we use this E_0 to normalize the values of corresponding E field results as discussed below (Supplementary Fig. 3 and 4).

It is noted that the electrolyte was assumed as a statically distributed medium instead of a dynamic one when calculating the electric field distribution for all cases. During the simulation of von Mises stress distributions on the dendrite protrusions (Fig. 3), the diameter of the simulated protrusion is assumed to be 5 μm , which is positioned at the center of the pinhole.

Supplementary Table 2. Resistivity and Young's modulus of different items for simulation

Item	Resistivity (Ω m)	Young Modulus (Pa)
Li	1×10^{-5}	4.9×10^9
PI film	1×10^{14}	/
Cu	/	2×10^{11}
Electrolyte	1	/
Separator	/	1×10^9

Revised text 2:

According to the above theory, when the practical D value in our experimental condition is larger than the order of magnitude of 10^{-7} ($\text{cm}^2 \text{s}^{-1}$), the electric field in the electrodeposition system would become a significant factor toward dendritic lithium growth. From literature^{8,9,10}, the Li^+ diffusion constant of 1 M LiTFSI in 1:1 (v/v) DOL:DME was at the order of 10^{-5} ($\text{cm}^2 \text{s}^{-1}$). Even considering the dimensional condition that the pinhole area is about 1/10 of the compartment area, the diffusion constant of lithium ion in the pinhole shall be at the order of magnitude of 10^{-6} ($\text{cm}^2 \text{s}^{-1}$), which is still larger than the calculated one from Chazaviel's model (10^{-7}). This indicates that the diffusion of electrolyte is at a fast-enough time scale to provide an electrochemically active surface inside the compartment at the range of current densities employed in the experiment (0.25 to 1.0 mA cm^{-2}), thus can maintain a stable electric current. Therefore, electric field becomes the dominant factor which determines the growth of lithium dendrite inside the compartments, according to the space-charge model in Chazaviel's theory¹. Consequently, the influence of potential gradient generated from the uneven anion depletion during a dynamic charging/discharge process is less significant, so that the electrolyte can be assumed to be a static medium (charge carrier) to simplify the simulation.

Updated reference:

1. Lin, Y. C., Li, M. & Wu, C. C. Simulation and experimental demonstration of the electric field assisted electroporation microchip for in vitro gene delivery enhancement. *Lab Chip* 4, 104-108 (2004).
2. Aryanfar, A. et al. Dynamics of lithium dendrite growth and inhibition: Pulse charging experiments and monte carlo calculations. *J. Phys. Chem. Lett.* 5, 1721-1726 (2014).
3. Ge, M., Rong, J., Fang, X. & Zhou, C. Porous doped silicon nanowires for lithium ion battery anode with long cycle life. *Nano Lett.* 12, 2318-2323 (2012).
4. Greve, L. & Fehrenbach, C. Mechanical testing and macro-mechanical finite element simulation of the deformation, fracture, and short circuit initiation of cylindrical lithium ion battery cells. *J. Power Sources* 214, 377-385 (2012).

8. Liu, W., Lin, D., Pei, A. & Cui, Y. Stabilizing lithium metal anodes by uniform li-ion flux distribution in nanochannel confinement. *J. Am. Chem. Soc.* **138**, 15443-15450 (2016).
9. Zheng, G. *et al.* Interconnected hollow carbon nanospheres for stable lithium metal anodes. *Nat. Nanotechnol.* **9**, 618-623 (2014).

Original Figures:

Supplementary Figure 3: Top-view distribution of normalized electric field strength inside the cavity with different lithium deposition volume %. (a)-(d) correspond to the cross-sections (thick black lines) right under the PI film, (e)-(h) are the cross-sections at the mid-plane of the cavity. The white color parts are Li deposition within the cross-section. For comparison purpose, (a) and (e) are the cases with 0% deposition and with no PI film. Cases (b) and (f) are the cases with 0% deposition. (c) and (g) are the cases with 20% deposition. (d) and (h) are the cases with 50% deposition.

Revised Figures:

Supplementary Figure 3: Simulation (top view) of normalized electric field magnitude distribution inside the compartment with different lithium deposition volume %. (a) Top view of simulation model. (b-c) Side view of simulation model. (d-g) Cross-sectional views of electric field magnitude distribution right under the PI film (thick black dash line in Supplementary Fig. 3b). (h-k) Cross-sectional views of electric field magnitude distribution at the mid-plane of the compartment (thick black dash line in Supplementary Fig. 3c). The white color parts are Li dendrites within the compartment. For comparison purpose, (d) and (h) are the cases with 0% deposition and with no PI film. (e) and (i) are the cases with 0% deposition. (f) and (j) are the cases with 20% deposition. (g) and (k) are the cases with 50% deposition.

(2) The connection between the modeling and experiments is lacking in the main paper and only discussed in the supplemental section. The manuscript would be improved by including details on the governing equations of the model and details of the Ansys models used.

Response: *Thanks for the comment. This constructive comment would significantly contribute to improve the quality of this paper. We have added the following discussions in the revised manuscript to reinforce the connection between the modeling and experiments, as depicted below.*

Revised text 1:

Simulation on electric field distribution and lithium plating/stripping behaviors in E-Cu

The distribution of electric field generated in P-Cu and E-Cu is schematically depicted in Fig. 2a. On the planar P-Cu, the direction of electric field exhibits a simple vertical pattern (perpendicular

to the separator). Whereas, the distribution of electric field generated inside E-Cu presents a unique lateral pattern, confirmed by numerical simulation using electrical conduction model as exhibited in Fig. 2b; in this pattern, the electric field propagates from the counter electrode, through the pinhole, and extends laterally to the Cu scaffold surface. This unique distribution pattern derives from the distortion effect of the top insulative PI layer on the electric field. Here, the electric field distribution is considered as one of the dominant factors (see supplementary information), which could modulate the growth behavior of lithium dendrite³⁵, as discussed below.

Along the distribution of electric field, plated Li metal primarily forms into small and mossy Li dendrites on the smooth surface of P-Cu due to limited electroactive sites. During the charging process, the subsequent dissolution of Li will result in many sharp ends and dead Li on the surface of P-Cu. Since Li metal is preferentially deposited along the sharp ends where local current density is dramatically increased⁵, larger Li dendrites and more dead Li will be evolved after repeated cycles (Fig. 2c). In contrast, owing to the existence of insulative PI film on E-Cu, Li metal is limited to deposit laterally inside the Cu scaffold and grows into Li dendrites. Even after cycling for a long time, Li dendrites will always be confined inside these hollow compartments as long as the cycling capacity is not exceeded (Fig. 2c). Here, the upper PI film in E-Cu can act as a physical barrier that shields the Li dendrites from protruding out of the pinholes in upper PI film. Despite some distortion effect on the electric field distribution within the compartment with the presence of this PI layer, electrochemical plating/stripping behavior of Li metal in E-Cu can still be observed, since the electric field can propagate into the compartment (see Supplementary Fig. 3 and 4 for discussion).

Once the cycling capacity approaches the limit capacity of E-Cu for effective Li storage, vertical Li dendrites will protrude out of the pinholes of upper PI film (extreme case). Even so, these protrusions are often long and curvy compared with short and sharp ones on P-Cu, meaning that they are mechanically much weaker than that on P-Cu, and therefore producing lower stress and being less probable to impale the separator. Accordingly, we simulated using structural stress analysis model, and observed an approximately 60% reduction of stress from protruded Li dendrites in E-Cu over P-Cu (Fig. 3). In these simulation cases, a vertical dendritic protrusion is positioned against the separator under different deposition capacities (Fig. 3a). When the compartment is filled up to the top PI film with Li dendrites (Fig. 3c-e), the stress on the separator is still significantly less than that of the control case (planar configuration, Fig. 3b). In another word, the predicted reduction of protrusion stress would further alleviate the safety problem of the lithium anodes based on E-Cu and enhance the structural integrity. It's worth mentioning that broken Li strips will stay inside the compartment, and thus Li metal can only grow along these broken strips until they are reconnected with newly formed lateral Li dendrite, which is quite different to the situation for P-Cu; in this case, the stress distribution in reconnected dendrites would be similar to the case without remnants.

Original text 2:

Numerical Simulation: A finite element analysis (FEA) simulation was performed on E-Cu@Li model with randomly generated cylindrical dendrite from the Cu scaffold in single compartment. The FEA package from ANSYS Inc. was used for simulation and post-processing.

Revised text 2:

Numerical Simulation: A finite element analysis (FEA) simulation was performed on bare E-Cu model and E-Cu@Li model to predict the compartment effects on the distribution of electric field as well as the growth of lithium dendrites inside the compartments. The E-Cu@Li model was established on single compartment with cylindrical dendrites inside, which were randomly generated using Monte Carlo method. The FEA package from ANSYS Inc. was used for simulation and post-processing.

(3) The manuscript contains a number of grammar error and many sentences are written in poor English making some statements hard to understand.

Response: Thanks for the comment. We have carefully reviewed the manuscript and asked a native speaker to polish the context. The language of the manuscript has been well improved.

Reviewer #2 (Remarks to the Author)

(1) The authors should compare their results to other, equally efficacious techniques and demonstrate that their technique offers benefits compared to them. One example to compare to is from Cui, attached.

Response: Thanks for the suggestion. We have supplemented additional discussion about the differences and advantages of our technology as compared with other people's works (including Cui's) in the revised manuscript.

As we mentioned in the introduction part of the manuscript, there have been a few strategies reported to deal with the lithium dendrites issue. The as-mentioned work done by Prof. Yi Cui's group¹ exemplified an effective method to suppress/delay the growth of lithium dendrite, which showed great promise in improving the safety level of the batteries. There are also many other works done by peer scientists, which are aimed to suppress/delay the growth of lithium dendrites as well. However, even though the risk of dendrite growth can be substantially inhibited by many of the available studies, to our authors' knowledge, there is no indication that the dendrite growth issue can be fully avoided;^{2,3} there is no statistical discussion about the probability of dendrite growth in the reported works either. Considering that in the conventional electrode structures, any case of lithium dendrite growth can potentially lead to fatal internal short-circuit incident, a rational design of electrode structure, which can eliminate short-circuit hazard when dendrites are already grown, is critically important^{3,4,5}.

To this end, we tentatively modulated the electric field distribution in the anode region, and successfully realized the lateral growth (namely parallel to the separator) of lithium dendrites by introducing a compartmented electrode structure. This unique electrode design renders the growth of lithium dendrites in lateral direction in a highly controllable and reproducible manner; which can also minimize dendrite stress toward separator, and thus further minimizing the hazard of short-circuits between electrodes, even when dendrites are already prevalently existed. On the other hand, the fabrication of this well-ordered micro-compartmented electrode structure is fully compatible to the current electronic circuit production processes, which is highly reliable and can be easily scaled up. In general, we consider this method would probably be the last defense line to prevent fatal internal short-circuit for the future commercial lithium metal batteries.

To be noted, this compartmented lithium anode exhibited a superior cycle life and a more stable voltage hysteresis (runs smoothly for over 150 cycles at 0.5 mA cm^{-2}) in Cu@Li//Li symmetrical batteries. When coupling this compartmented anode with commercialized LiFePO_4 cathode, the assembled full cell can run for 250 cycles with a capacity retention of 100.7% and a Coulombic efficiency (CE) of 99.5 at 1C.

Finally, this technology is compatible with most of the state-of-the-art dendrite-growth-delay/suppression technologies, and can be further improved with the development of microfabrication and computational techniques.

Reference:

1. Zheng, G. et al. Interconnected hollow carbon nanospheres for stable lithium metal anodes. *Nat. Nanotechnol.* **9**, 618-623 (2014).
2. Brissot, C., Rosso, M., Chazalviel, J. N., Baudry, P. & Lascaud, S. In situ study of dendritic growth in lithium/peo-salt/lithium cells. *Electrochimica Acta* **43(10)**, 1569-1574 (1998).
3. Cheng, X. B., Zhang, R., Zhao, C. Z. & Zhang, Q. Toward safe lithium metal anode in rechargeable batteries: A review. *Chem. Rev.* **117**, 10403-10473 (2017).
4. Sun, Y., Liu, N. & Cui, Y. Promises and challenges of nanomaterials for lithium-based rechargeable batteries. *Nat. Energy* **1**, 16071 (2016).
5. Lin, D., Liu, Y. & Cui, Y. Reviving the lithium metal anode for high-energy batteries. *Nat. Nanotechnol.* **12**, 194-206 (2017).

Revised text 1:

These achievements have provided elaborate insights into the feasibility of effective dendrite suppression via stabilization and homogenization of solid-electrolyte-interphase (SEI) layer or accommodation of electrodeposited Li metal, and would promote the commercialization of metallic lithium anode in Li-metal-based secondary batteries. Whereas, **there are rare discussions on the extreme situation when the control in suppressing/delaying the dendrites is failed in these strategies because** the emergence of Li dendrites can't be completely avoided during prolonged cycling³, especially when batteries are operated at high current densities, in overcharge ultimate, or at low operation temperatures^{26, 27}.

Revised text 2:

Herein, we develop a scalable technology with photolithographic-level conformity for the fabrication of polyimide (PI)-clad copper grid current collectors (E-Cu) for Li metal anodes, where the electric field presents a lateral pattern inside E-Cu and thus guides the Li dendrites to grow laterally within the interior Cu scaffold. **Instead of suppressing/delaying the dendritic growth, this technology is dedicated to regulate the dendrite growth direction parallel to the separator so that the batteries can still work safely even when dendrites are already massively existed.** All the processes involved, including hot lamination, laser ablation and alkaline etching (as schematically illustrated in Fig. 1a), have been widely used in the fields of electronic and semiconductor industry for more than half a century^{32, 33, 34}, which can ensure the highest conformity level.

Revised text 3:

In summary, this manuscript demonstrates that guiding Li dendrites to grow laterally in compartmented micro-electrodes is an effective way to manage the safety issue of Li anodes, **which is different to the widely adopted strategies in suppressing/delaying the dendritic growth.** The model structure can effectively change the electric field distribution and make accommodation to the plated Li metal.

Revised text 4

Besides, the excellent compatibility between this compartmented electrode structure with industrially-available fabrication techniques, including hot lamination, laser ablation and alkaline etching, also renders this technology with unprecedented conformity and reliability. Therefore, it would be a critical step towards large scale manufacturing of Li metal anode based batteries. **This unique strategy is a first attempt to deal with the extreme situation when lithium dendrites have massively presented via manipulating the electric field distribution and growth dynamics of Li dendrites, which provides new insights into the unwelcomed dendrite-growth issue, and will inspire technological development of other metal anodes in rechargeable systems.**

(2) Otherwise, I don't understand the electric field calculations. They appear to have been done assuming no electrolyte, which is unrealistic. In fact, much of the potential gradients will occur through the electric double layer. Is there something I am missing?

Response: Thanks for the question. Indeed, the electrolyte was taken into consideration as a dielectric medium, when conducting the numerical calculations in this work. Electrolyte acts as a charge carrier, coupling with electron carrier (current collector, namely Cu scaffold in this work), which are essential in building the electric field in electrochemical systems. During the simulation, the electrolyte was assumed as a statically distributed medium instead of a dynamic one based on the fact that the electric field would be the dominant factor to the growth of lithium dendrite when sufficient solute concentration is available for electrochemical reaction, according to the space-charge model in Chazavie's theory¹ (as discussed in detail in the revised supporting information). Based on this reasonable assumption, we considered that the influence of potential gradient was less significant, where the potential gradient was generated from the uneven anion depletion during a dynamic charge/discharge process. Consequently, in this manuscript, we assume the electrolyte as a static medium (charge carrier) so as to simplify the simulation for all cases. We also note that, taking the dynamic electric field and concentration field into consideration at the same time would help to improve calculation accuracy a little bit; but to our authors' knowledge, the related theoretical models have never been reported yet. The related model set-up and calculation works are still ongoing in our group.

Finally, we have updated the corresponding sections by elucidating in more details about the simulation setup, as shown in the revised supporting information.

Reference :

1. Chazalviel, J. N. Electrochemical aspects of the generation of ramified metallic electrodeposits. *Phys. Rev. A* **42**, 7355-7367 (1990).

Revised text 1:

It is noted that the electrolyte was assumed as a statically distributed medium instead of a dynamic one when calculating the electric field for all cases. During the simulation of von Mises stress distributions on the dendrite protrusions (Fig. 3), the diameter of the simulated protrusion is assumed to be 5 μm , which is positioned at the center of the pinhole.

Revised text 2:

This indicates that the diffusion of electrolyte is at a fast-enough time scale to provide an electrochemically active surface inside the compartment at the range of current densities employed in the experiment (0.25 to 1.0 mA cm^{-2}), thus sustaining the electric current. Therefore, electric field becomes the dominant factor which can determine the growth of lithium dendrite inside the compartments, according to the space-charge model in Chazavie's theory. Consequently, the influence of potential gradient generated from the uneven anion depletion during a dynamic charge/discharge process is less significant, so that the electrolyte can be assumed to be a static medium (charge carrier) to simplify the simulation.

(3) Because this paper presents a wealth of interesting data and analysis, it would be *my personal preference* to see a longer version of this paper. For example, it would be nice if some of the 16 Supplementary Figures could appear in the text, where more people would see them.

Response: *Thanks for the comment, which could well contribute to improve the quality of this paper. Here we have added three Supplementary Figures, including the cross-view SEM image of E-Cu (Fig. 1e), porosity analysis result for E-Cu with mercury porosimetry (Fig. 1f), and the simulation of von Mises stress distributions on the dendrite protrusions (Fig. 3), in the revised manuscript. In addition, the corresponding description are also updated in the revised manuscript.*

Original Figures:

Figure 1

Revised Figures:

Figure 1 Preparation and characterization for E-Cu. (a) Schematic of fabrication process of E-Cu. (b) Photograph of the as-obtained E-Cu current collector. (c) Typical overview image of E-Cu after peeling off the coated PI films. (d) Magnified image of the edge part of copper scaffold from (c). Scale bars in (b)-(d) are 1 cm, 100 μm , and 1 μm , respectively. (e) Typical cross-view image of E-Cu after peeling off the surface PI films. (f) Cumulative pore volume of E-Cu by

mercury porosimetry. Herein, the diameter of interior compartments is controlled to be $\sim 150 \mu\text{m}$, which was in accordance to the porosity analysis result ($148 \mu\text{m}$), as marked in red color. The weight of E-Cu tested here was 0.3141 g and the total surface area for all compartments was 9 cm^2 ; thus, the effective pore volume of E-Cu is $1.88 \times 10^{-3} \text{ cm}^3 \text{ cm}^{-2}$.

Figure 3 Simulation on von Mises stress distribution of the dendritic protrusions at different deposition heights. A vertical protrusion is grown from the lower deposition to the separator in all cases. It is located at the center of the XY-plane. (a) 3D and cross-sectional view of E-Cu@Li model. (b) Distribution of protrusion stress in the control case with two-plate electrodes. (c-e) Distribution of protrusion stress in the E-Cu base cases with dendrites layer reaching the (c) 95%, (d) 75% and (e) 50% height of the compartment, respectively. On the right hand corners in Fig. 3b-e, the relative area-average von Mises stress from the protrusion toward the separator is indicated.

Reviewers' comments:

Reviewer #1 (Remarks to the Author):

My comments have been addressed. However, the English is still poor with incorrect grammar and awkward sentences. The manuscript would benefit from further editing by a native English speaker.

Reviewer #2 (Remarks to the Author):

I am satisfied that the authors have responded adequately to my comments. I recommend publication

RESPONSE TO REFEREE'S LETTER

We appreciate the reviewer's comments. Detailed responses are as follows.

Reviewer #1 (Remarks to the Author)

(1) My comments have been addressed. However, the English is still poor with incorrect grammar and awkward sentences. The manuscript would benefit from further editing by a native English speaker.

Response: *Thanks for the comment. We have carefully reviewed both the manuscript and supplementary information, and asked a few native speakers to polish the language to improve its accuracy, clarity and readability. All the changes have been highlighted in the revised manuscript and supplementary information. Below are some examples before and after revision.*

Original text 1:

Whereas, there are rare discussions on the extreme situation when the control in suppressing/delaying the dendrites is lost in these strategies because the emergence of Li dendrites can't be completely avoided during prolonged cycling³, especially when batteries are operated at high current densities, in overcharge ultimate, or at low operation temperatures^{26, 27}.

Revised text 1:

Whereas, **the extreme situation when controlled dendrite suppression/delay is lost in these strategies has been rarely discussed** because the emergence of Li dendrites cannot be completely avoided during prolonged cycling³, especially when batteries are operated at high current densities, in overcharge ultimate, or at low operation temperatures^{26, 27}.

Original text 2:

On the other hand, the electro-deposition/dissolution behaviors of Li metal and corresponding influence factors are intrinsically complicated, rendering it hardly predictable and extremely difficult to be managed with available technologies.

Revised text 2:

On the other hand, the electrodeposition/dissolution behaviours of Li metal and the corresponding **influencing factors** are intrinsically complicated, rendering **the control over these behaviours** hardly predictable and extremely difficult to be managed with available technologies.

Original text 3:

Accordingly, we simulated using structural stress analysis model, and observed an approximately 60% reduction of stress from protruded Li dendrites in E-Cu over P-Cu (Fig. 3).

Revised text 3:

Accordingly, we simulated **this stress** using a structural stress analysis model and observed an

approximately 60% reduced stress from the protruded Li dendrites in E-Cu as compared to that in P-Cu (Fig. 3).

Original text 4:

In addition, several dendrites-induced short circuits occurred in P-Cu based Li anode during cycling, which can be observed from the voltage profile where voltage abruptly swooped from high potentials to lower ones as indicated in Fig. 6c.

Revised text 4:

In addition, several dendrites-induced short circuits occurred in the P-Cu based Li anode during cycling, which can be observed from abrupt voltage drops from high potentials to lower potentials in the voltage profile, as indicated in Fig. 6c.

Original text 5:

In a sandwich cell structure, the distribution of electric field predominately presents a vertical pattern; thus once Li dendrite forms, it will grow vertically towards separator and cathode^{3, 4, 29, 30}, and eventually could impale the separator and cause internal short circuit of the battery³.

Revised text 5:

In a sandwich cell structure, the electric field is predominately distributed in a vertical pattern; thus, upon Li dendrite formation, the dendrite will grow vertically towards the separator and cathode^{3, 4, 29, 30}, and eventually could impale the separator and cause an internal short circuit in the battery³.